# Anti-Inflammatory Effect of Chloroform Fraction of *Pyrus Ussuriensis Maxim.* Leaf Extract on 2, 4-Dinitrochlorobenzene-Induced Atopic Dermatitis in nc/nga Mice

**DOI:** 10.3390/nu11020276

**Published:** 2019-01-27

**Authors:** KyoHee Cho, Min Cheol Kang, Amna Parveen, Silvia Yumnam, Sun Yeou Kim

**Affiliations:** 1College of Pharmacy, Gachon University, 191, Hambakmoero, Yeonsu-gu, Incheon 21936, Korea; kcho3138@gmail.com (K.C.); mincjf07@gmail.com (M.C.K.); amnaparvin@gmail.com (A.P.); 2Gachon Institute of Pharmaceutical Science, Gachon University, 191, Hambakmoe-ro, Yeonsu-gu, Incheon 21936, Korea; 3Department of Pharmacognosy, Faculty of Pharmaceutical Science, Government College University, Faisalabad 38000, Pakistan; 4Gachon Medical Research Institute, Gil Medical Center, Incheon 21565, Korea

**Keywords:** atopic dermatitis, DNCB-induced-NC/Nga mice, astragalin, TNF-α/IFNγ, IL-6, PULC

## Abstract

*Pyrus ussuriensis Maxim*, a pear commonly known as “Sandolbae” in Korea, is used as a traditional herbal medicine for asthma, cough, and fever in Korea, China, and Japan. *P. ussuriensis Maxim* leaves (PUL) have therapeutic effects on atopic dermatitis (AD). However, there are no reports on the efficacy of specific components of PUL. In the present study, activity-guided isolation of PUL was used to determine the compounds with potent activity. Astragalin was identified as the major component of the chloroform-soluble fraction of PUL (PULC) using High-performance liquid chromatography (HPLC) analysis. Astragalin and PULC were tested in vitro and in vivo for their effects against AD. PULC and astragalin dose-dependently inhibited the production of nitric oxide (NO) in mouse macrophage (RAW 264.7) cells, and interleukin (IL)-6 and IL-1β in tumor necrosis factor (TNF-α)/interferon γ (IFNγ) induced HaCaT cells. In the AD mice model, PULC and astragalin application significantly reduced dermatitis severity, scratching behavior, and trans-epidermal water loss (TEWL) when compared to that of 2, 4-dinitrochlorobenzene-treated NC/Nga mice. Additionally, they normalized skin barrier function by decreasing immunoglobulin E (IgE) levels in the serum. Filaggrin and involucrin protein levels were normalized by PULC treatment in HaCaT cells and skin lesions. These results indicate that PULC and astragalin ameliorate AD-like symptoms by alleviating both pro-inflammatory cytokines and immune stimuli in vitro and in vivo in animal models. Therefore, PULC and astragalin might be effective therapeutic agents for the treatment of AD.

## 1. Introduction

Atopic dermatitis (AD), a chronic inflammatory skin disease, is due to the complex linkage between environmental and genetic mechanisms [1]. It is a common disease affecting infants and children with an estimated incidence of 15–30% in children and 2–10% in adults [1]. Changes in lipid composition in the epidermis facilitate the permeability of allergens and irritate the skin, which cause an overproduction of cytokines and chemokines and ultimately maintain and promote inflammation. AD causes a complex activation of immunologic and inflammatory pathways. In AD, cytokines like TNFα and IL-1β binds to vascular endothelial receptors, which activates the NFκB pathways [2]. This is followed by stimulation of inflammatory cytokines and chemokines. The skin barrier function is also highly impaired in AD. Additionally, AD patients show elevated levels of immunoglobulin E (IgE) and penetration of inflammatory cells such as macrophages, eosinophils, and mast cells [3,4,5]. Thus, AD is a chronic inflammatory skin disease, and some of the characteristic clinical features are dry itchy skin, pruritis, and eczema with increased trans-epidermal water loss. Nitric oxide (NO), interleukin (IL)-1β, and IL-6 are the most highly expressed cytokines related to the immune response in the pathogenesis of AD [6,7,8,9,10]. Therefore, for the treatment of AD, potential targets are still needed for therapeutics. The commonly used treatment for AD is the local or systemic application of corticosteroids, emollients, immunosuppressants, and antihistamines [11,12]. Even though these drugs are popularly used for the treatment of AD, there is an urgent need for therapeutics which specifically target AD. 

The associated skin changes that occur in human AD are induced in NC/Nga mice through exposure to several environmental aeroallergens [13,14]. In line with this view, 2, 4-dinitrochlorobenzene (DNCB) is known to cause stable AD-like skin disease in NC/Nga mice [15,16]. The skin changes were characterized by excessive eczema with constant scratching erythema, skin dryness, and hemorrhages [16,17]. The NC/Nga mice also exhibited increased levels of IgE and DNCB-induced hypersensitivity due to the development of immune responses controlled by T cells [16].

Currently, various herbal medicines are used for the prevention of AD and asthma diseases [18,19]. *Pyrus ussuriensis* is the major cultivated species of pear and is widely distributed in China, Korea, and Japan. It is known for its medicinal properties including anti-inflammatory, antioxidant, antitussive, and diuretic effects [20]. In Korea, it is a part of traditional medicine mentioned in the ancient traditional medicine book “Dongui Bogam”. Previous pharmacological studies on *P. ussuriensis* reported that its leaf extract has antioxidant [20] and anti-inflammatory effects that may be useful for the treatment of AD. In a previous study, we demonstrated that *P. ussuriensis* leaf extract inhibited symptoms of AD in the DNCB-induced NC/Nga mouse model [21]. In the present study, the bioactivity-guided fractionation of *P. ussuriensis Maxim* leaves (PUL) was used to determine the compounds with potent activity and investigate the therapeutic effects of *P. ussuriensis* and its components in vitro in macrophages and keratinocytes and in vivo in the DNCB-induced AD-like skin lesion model using NC/Nga mice.

## 2. Material and Methods 

### 2.1. Fractionation and Isolation of Pul Extract 

*P. ussuriensis Maxim.* leaf extract was prepared as described by Cho et al. [21]. The liquid–liquid partition of the 70% EtOH total extract (PULT, 525 g) was fractionated with hexane (PULH), chloroform (PULC), ethyl acetate (PULE), n-butanol (PULB), and water (PULW). The solvents were then evaporated and freeze-dried to obtain 55, 23, 41, 5, and 287 g of respective samples. Further, the chloroform fraction was separated using stepwise gradient elution of ethyl acetate-methanol from [8:2] to [0:10] in a silica gel column chromatography which results in 1–7 sub-fractions. The collected samples were then analyzed by thin-layer chromatography (TLC), and fractions showing similar TLC patterns [ethyl acetate: methanol: water, (100:13.5:10)] were combined. As a result, 7 distinct sub-fractions were identified and named sequentially from “C1” to “C7”. The C2 sub-fraction was further eluted using Sephadex LH-20 open column with a stepwise gradient elution of methanol-water from [10:90] to [100:0]. Four distinct sub-sub fractions were identified and named sequentially from “a” to “d”, respectively. The process of the preparation of the extract is illustrated in Figure 1.

### 2.2. HPLC Analysis of PULC

As previously described by Cho et.al., [21] HPLC analysis of PULC was carried out using the Waters system (Waters Corp., Milford, MA, USA) with a separation module (e2695) equipped with a photodiode array detector. Reverse phase chromatography was performed using a C18 column (250, 4.6 mm × 5 µm). Mobile phase consists of methanol (solvent A) and 0.1% phosphoric acid (solvent B); an isocratic elution (0–30 min, 40% A) was used at a flow rate of 1 mL/min. The column temperature was set at 30 °C and 10 µL of the sample was injected. Astragalin (Sigma-Aldrich Chemicals, USA; purity ≥95 %) was used as the standard compound at concentrations of 0.1, 1, and 10 μg/mL to identify the peaks. The ultraviolet (UV) absorption spectra was recorded at 254 nm. 

### 2.3. Cell Culture

RAW 264.7 and human HaCaT cells were purchased from the Korean Cell Line Bank (Seoul, Korea) and were maintained in Dulbecco’s modified Eagle’s medium containing 1% of penicillin (1 × 10^5^ U/L), streptomycin (100 mg/mL), and 10% of fetal bovine serum at 37 °C in a humidified CO_2_ (5%) incubator. 

### 2.4. MTT Assay 

Cell viability was evaluated using 3-(4,5-Dimethylthiazol-2-yl)-2,5-diphenyltetrazolium bromide (MTT) assay according to Hwang et al. [22]. RAW 264.7 and HaCaT cells were seeded overnight in a 96-well plate and treated with various concentrations of PUL fractions and sub-fractions with or without lipopolysaccharide (LPS) and TNFα/IFN-γ, respectively. Following 24 h treatment, the media were replaced with 0.5 mg/mL of MTT solution. After 1 h of incubation, the MTT solution was removed and 200 µL of dimethyl sulfoxide (DMSO) was added to each well. The optical density was recorded at 570 nm using a plate reader. 

### 2.5. Assessment of NO Production in RAW 264.7 Cells 

Effect PUL sub-fractions on LPS-induced NO production in RAW 264.7 cells were performed according to Han et al. with some adjustments [23]. RAW 264.7 cells were seeded at a density of 4 × 10^4^ cells/well in a 96-well plate and were pretreated with PUL sub-fractions, followed by LPS activation for 24 h. Using Griess reagent, NO levels were measured in the culture media. Briefly, 50 µL Griess reagent was added to 50 µL conditioned medium and the absorbance was measured at 570 nm. 

### 2.6. Evaluation of IL-6 and IL-1β in TNF-α/ IFN-γ-Induced HaCaT Cells

IL-6 and IL-1β secretion was measured using enzyme-linked immunosorbent assay (ELISA). HaCaT cells were seeded at a density of 2 × 10^5^ cells/well in a 24-well plate. When the cells reached a confluency of 70–80%, they were stimulated with 10 ng/mL of TNFα or IFN-γ, and with PULC and its sub-sub fractions. After 24 h, the supernatants were collected and IL-6 and IL-1β secretion was evaluated using the respective ELISA kit (R&D systems, Minneapolis, MN, USA). 

### 2.7. DNCB-Induced AD model

NC/Nga female mice (4 weeks old) were purchased from Orient Bio (Gyeonggi-do, Korea) and were maintained in a pathogen-free environment at a temperature of 22 ± 2 °C and a 12-h light/dark cycle. The mice were randomly divided into four groups (9 per group): NOR, normal control vehicle (NOR, CON (DNCB + vehicle), DEX (DNCB + positive control treated with dexamethasone 0.1%), PULC, and DNCB + PULC 0.5%. To induce AD-like skin lesions. The skin of the animals was shaved on day 6; 200 µL of DNCB (1%) was dissolved in acetone + olive oil mixture (3:1, v/v) and then applied to the dorsal skin on day 7. For five weeks, 200 μL of each sample was applied on the back of both the ears twice weekly. All the five groups were then treated according to the treatment group. Dorsal skin tissues obtained from the mice were subjected to histological and western blot analysis. All procedures were performed in accordance with the guidelines for the Care and Use of Laboratory Animals (LCDI-2017-0038) approved by the Institutional Animal Care and Use Committee of Gachon University (Incheon, Korea).

### 2.8. Measurement of Trans-Epidermal Water Loss (TEWL), Hydration, and Erythema 

Dermalab Combo system (C40000.03-189, Cortex Technology, Hadsund, Denmark) was used to measure TEWL, hydration and erythema of the epidermal layer of the skin of Nc/Nga mice. At approximately 21–22 °C and humidity of 50–55%, the measurements were made by placing a probe on the mice’s skin that was stabilized for approximately 30 s; data were recorded in g/m^2^/h. The measurements were performed from week 7 of the experiment. 

### 2.9. Evaluation of Skin Lesions and Dermatitis Score

Once a week, dermatitis severity score was checked as previously described [24]. The severity of dermatitis was calculated according to four symptoms: edema, erythema/hemorrhage, erosion/excoriation, and dryness. 0 score was recorded for none, 1 for mild, 2 for moderate, and 3 for severe conditions. The clinical severity of dermatitis was the sum of the overall individual scores.

### 2.10. Scratching Behavior Evaluation

On the final day of the experiment, scratching behavior of each mice was recorded using a digital camera (Coolpix A300, Nikon imaging Korea, Seoul, Korea) as described [24] by counting the number of hindlimb scratch bouts during 20 min. 

### 2.11. Serum IgE 

Measurement of serum IgE level was analyzed as previously described [25]. On the last day of the experiment, blood and dorsal skin were collected and kept at −80°C until use. Serum IgE levels were measured using ELISA kit (R&D System).

### 2.12. Histological Examination

For histological examination, the dorsal skin of each mouse collected on the last day of the experiment was fixed with 10% neutral-buffered formalin and embedded in paraffin. Skin sections of 4 µm thickness were cut and transferred to slides. Before examination at 100× magnification, hematoxylin and eosin (H&E), IL-1β, and toluidine blue stains were used to stain the deparaffinized skin sections. The stained skin was observed under a Nikon Eclipse 80i Microscope (Tokyo, Japan) at 100× magnification. 

### 2.13. Western Blotting

Cell lysates and/or dorsal skin tissues were lysed overnight in PRO-PREP™ protein extraction solution (Intron Biotechnology, Inc.; Seoul, Korea) at 4 °C. The lysates were centrifuged at 11,000× g for 30 min at 4 °C. The supernatants were collected, and protein concentrations were determined using Bio-Rad protein assay reagent (Bio-Rad Laboratories, Inc.) according to the manufacturer’s protocol. Proteins (20 μg) were separated using sodium dodecyl sulfate polyacrylamide gel electrophoresis (SDS-PAGE) 12% and blotted onto a polyvinylidene fluoride membrane (PVDF). The membranes were blocked for 1 h in 5% of skim milk at room temperature, followed by overnight incubation with primary antibodies against filaggrin, involucrin, and glyceraldehyde-3-phosphate dehydrogenase (GAPDH) at 4 °C. After incubation with the respective horseradish peroxidase-conjugated secondary antibodies, the blots were developed via enhanced chemiluminescence (GE Healthcare Life Sciences, Chalfont, UK). Densitometric analysis was performed using Bio-Rad Quantity One software version 4.3.0 (Bio-RadLaboratories Inc, Hercules, California, USA).

### 2.14. Statistical Analysis

The data are expressed as the mean ± standard error of the mean (SEM) using GraphPad Prism 5.0 (GraphPad Software Inc., San Diego, CA, USA). One-way analysis of variance (ANOVA) followed by Tukey’s honest significance test was used to evaluate the statistical comparison between control and various groups. The differences were considered statistically significant at a *p*- value (<0.05 *, <0.01 **, and <0.001 ***).

## 3. Results

### 3.1. Measurement of Astragalin Content from PULC by HPLC Analysis

HPLC-UV analysis at 254 nm was used to determine the components of PULC. Astragalin was found to be the major constituent of PULC. Its retention time (Rt: 20.53 ± 0.28 min) matched with its standard. Content analysis indicated that PULC contained 0.03 ± 0.00 μg/100μg of astragalin (Table 1).

### 3.2. Bioactivity-Guided Fractionation of PULC Based on Its Anti-Inflammatory Effects

As shown in Figure 2, PUL fractions attenuated cell death and decreased NO production as compared to the negative control in RAW 264.7 cells. PULC at a dose of 10 µg/mL significantly reduced NO production to 14.26% when compared to negative control. Thus, the chloroform-soluble fraction of PULC was further sub-fractionated in an open column chromatography, which resulted in seven sub-fractions (C1–C7). Bioactivity-guided fractionation of PULC fractions were performed for NO production in RAW 264.7 cells. Sub-fraction C2 at a dose of 20 µg/mL also lowered NO production to 80.02% compared to the negative control (Figure 2D). Thus, this sub-fraction was further fractionated and studied for other anti-inflammatory effects.

### 3.3. PULC Reduced IL-6 and IL-1β Levels in TNF-α/IFN-γ Treated HaCaT Cells

An inhibitory effect of PULC and C2 sub-sub fractions on IL-6 and IL-1β production was observed in TNF-α/ IFN-*γ* treated HaCaT cells. PULC and sub-sub fraction B both at a dose of 10 µg/mL lowered IL-6 levels to 67.60% and 59.41% and IL-1β levels to 50.43% and 45.68%, respectively compared to negative control. Also, 100 µg/mL astragalin lowered IL-6 levels to 60.44% and IL-1β levels to 43.60% compared to TNF-α/ IFN-*γ* treated group (CON). The inhibitory effect of inflammatory cytokines was dose-dependent and no cytotoxicity was detected (Figure 3A–I).

### 3.4. Effect of PULC on Nc/Nga Mice 

TEWL and erythema in the epidermal skin layer of NC/Nga mice and skin hydration were evaluated in DNCB-treated Nc/Nga mice between 5th and 11th week. On the final day of the experiment, skin hydration was decreased in DNCB-induced NC/Nga mice (61.02 ± 2.15 g/m^2^/h) as compared with that in the normal control vehicle treated group (19.82 ± 2.63 g/m^2^/h). A decrease in skin TEWL was observed in the dexamethasone (0.1%) (26.72 ± 4.68 g/m^2^/h) and PULC (0.5%) (39.85 ± 5.63 g/m2/h) treated groups. By week 9, erythema level was increased in DNCB-treated NC/Nga mice (12.47 ± 1.58 RI) compared with the normal control group (7.58 ± 0.80 RI) while the erythema level was reduced in dexamethasone (0.1%) (11.85 ± 0.41 RI), and PULC (0.5%) (9.32 ± 0.68 RI) application in DNCB-treated mice (Figure 4B). 

### 3.5. PULC Alleviates DNCB-Induced AD-Like Symptoms in NC/Nga Mice

To investigate the effect of PULC on DNCB-induced AD-like symptoms in NC/Nga mice, we imaged skin lesions and dermatitis scores were evaluated. On the final day of the experiment, dorsal skins of DNCB-treated NC/Nga mice showed severe erythema, erosion, and dryness. Treatment with dexamethasone (0.1%) and PULC decreased AD-like symptoms in DNCB-treated NC/Nga mice. The increased dermatitis score in DNCB-treated NC/Nga mice (14.87 ± 2.22) was significantly decreased in the dexamethasone (0.1%) (10.5 ± 1.64) and PULC (0.5%) (8.00 ± 1.28) treated groups. PULC treatment decreased the dermatitis score when compared with CON group (46.21%) (Figure 4C).

### 3.6. PULC Alleviates DNCB-Induced Scratching Behavior in NC/Nga Mice

To evaluate the antipruritic effect of PULC on DNCB-induced NC/Nga mice, scratching behaviors were monitored, and quantified the time spent by every single mouse rubbing their dorsal skin and ears with their hind paws as scratching time. An increased scratching time was observed on the final day of the experiment in DNCB-treated NC/Nga mice (432.21 ± 106.80 s) when compared with that of the normal controls (82.46 ± 18.89 s). The scratching time was significantly reduced in dexamethasone (0.1%) (250.36 ± 64.82 s) and PULC (0.5%) (83.63 ± 39.24 s) treated groups. Also, 0.5% PULC significantly decreased scratching time when compared with that of controls (8.65%) (Figure 4D).

### 3.7. PULC Decreased Serum IgE Levels in DNCB-Induced NC/Nga Mice

To evaluate the serum IgE levels in DNCB induced NC/Nga mice, a blood sample was collected on the last day of the experiment. It was observed that serum IgE levels were increased in DNCB-treated NC/Nga mice (17472.2 ± 1718.86 ng/mL) when compared with that of the control group (2000±961.04 ng/mL). These increased serum IgE levels were significantly decreased in dexamethasone (0.1%) (9527.78 ± 73.49 ng/mL) and PULC (0.5%) (10861.1 ± 1209.85 ng/mL) treated groups (Figure 4E). Also, 0.5% PULC treatment significantly decreased the IgE level when compared with the control group (37.83%).

### 3.8. Histological Examination

The skin of DNCB-treated NC/Nga mice became swollen and exhibited epidermal hypertrophy. In contrast, reduced infiltration of inflammatory cells and less severe epidermal hypertrophy were observed in the skin of mice treated with dexamethasone (0.1%) and PULC (0.5%) compared to those in the skin of DNCB-treated NC/Nga mice. H&E staining showed increased epidermal thickness by DNCB treatment (148.6 ± 6.72 µm), whereas a significantly reduced thickness was measured after treatment with dexamethasone (0.1%) (64 ± 4.53 µm) and PULC (0.5%) (45.6 ± 3.76 µm) in DNCB-treated mice (Figure 4Fa,G). IL-1β expression also decreased by PULC treatment in DNCB-induced skin lesions (Figure 4Fb,H). An elevated serum IgE level is associated with mast cell infiltration. Therefore, we examined the number of mast cells by toluidine blue staining in the dermal area. DNCB increased 9-fold (165.6 ± 8.8 cells) compared with that in the normal control group, whereas PULC treatment significantly reduced this number compared with that in the control group (59.6 ± 7.6 cells) (Figure 4Fc,I).

### 3.9. Effects of PULC and Astragalin on Protein Expressions of Filaggrin and Involucrin

To investigate whether PULC treatment could alter the skin moisturizing response, we assessed the effect of PULC^1^ on protein levels of filaggrin and involucrin in dorsal skin lesions of the DNCB-treated NC/Nga mice (Figure 5A, a–c). Moreover, the effects of astragalin on protein levels of filaggrin and involucrin in TNF-α- and IFN-γ-treated HaCaT cells were assessed. As a result, filaggrin and involucrin protein expression levels were significantly upregulated in groups treated with 100 µg/mL PULC^2^ (filaggrin: 0.43%, involucrin: 59.36%) and astragalin (filaggrin: 89.88%, involucrin: 40.67%) compared to those in the CON^2^ (Figure 5B, a–c).

## 4. Discussion

AD, characterized by pruritic eczematous skin lesions, is one of the most common inflammatory skin diseases with a broad spectrum of clinical skin phenotypes [26]. The pathology of AD skin is characterized by epidermal intercellular edema and skin barrier dysfunction resulting in increased TEWL and marked mononuclear cell count [27]. AD patients usually use topical emollients and moisturizers to hydrate the skin. 

Currently, various herbal medicines have been used in the prevention of AD and asthma [18,19]. In our previous study, we demonstrated that *P. ussuriensis* leaf extract inhibited AD-like symptoms in a DNCB-induced NC/Nga mouse model [21]. In the present study, bioactivity-guided fractionation of PUL has been used to evaluate the anti-AD activity in both in vitro and in vivo models. PULC and C2 sub-fraction significantly reduced NO production compared to negative controls dose-dependently with no cellular toxicity. NO is a known pro-inflammatory mediator and is produced excessively in inflammatory conditions [28]. AD being a chronic inflammatory skin disease is characterized by overexpression of various pro-inflammatory cytokines. During the inflammatory process, primary inflammatory cytokines TNF-α and IL-1β activate the production of secondary inflammatory cytokines like IL-6, IL-8, IL-12, IL-15, IL-17, and IL-18 [29]. TNF-α and IFN-γ stimulation in keratinocytes has been reported to induce inflammatory cytokines and chemokines [30,31]. Thus, inhibiting these inflammatory cytokines is an important target in reducing inflammation. In this study, IL-6 and IL-1β protein expression was significantly inhibited by PULC and their sub-fractions in TNF-α/IFN-γ-treated HaCaT cells. Moreover, astragalin, which was identified as the active compound in PULC sub-fraction C2, dose-dependently inhibited the TNF-α/IFN-γ-induced inflammation in HaCaT cells by inhibiting TNF-α/IFN-γ-induced IL-6 and IL-1β. These results suggest that PULC and C2 sub-fractions have anti-inflammatory properties.

To further elucidate the anti-AD role of PULC, PULC was topically applied in the DNCB-induced AD Nc/Nga mouse model. The Nc/Nga mouse AD model is considered to be clinically similar to human AD [32]. TEWL and serum IgE are high in patients with skin barrier dysfunction and are important biomarkers in AD [33]. Pro-inflammatory cytokines are also believed to increase IgE levels [34]. In our study, we observed that PULC decreased DNCB-induced erythema, dryness, and scratching frequency in Nc/Nga mice. Furthermore, PULC significantly reduced skin TEWL and serum IgE levels in DNCB-treated Nc/Nga mice. A dermatitis score of 14.87 ± 2.22 in DNCB-treated Nc/Nga mice was significantly reduced to 8.00 ± 1.28 in the PULC-treated group. These results suggest that PULC suppressed the AD skin symptoms induced by DNCB in the Nc/Nga mouse model.

Histological analysis of AD patients shows acute eczematous lesions with hypertrophy, hyperkeratosis, and perivascular infiltration of mast cells and lymphocytes [34,35]. In this study, histological evaluation of DNCB-treated Nc/Nga mice skin showed severe hypertrophy and high infiltration of inflammatory cells, which were effectively reduced by PULC treatment (Figure 5F). 

As skin barrier function is compromised in AD, filaggrin and involucrin, which play a major role in the formation of the skin barrier, are suppressed in AD. Filaggrin (a filament aggregation protein) plays a critical role in the differentiation of skin keratinocytes in the stratum granulosum [5]. Decreased expression of filaggrin in the skin and loss-of-function mutations in the filaggrin gene have been described in AD. Mutations in the filaggrin gene can lead to downstream immunologic activation, leading to the synthesis and secretion of specific IgE antibodies against allergens, causing abnormalities in the skin barrier [36]. A recent study also reported that DNCB-induced AD downregulates filaggrin and involucrin expression [37]. We found that DNCB-induced downregulation of filaggrin and involucrin was significantly increased in PULC treatment. These results correlate with the reduced TEWL and IgE in skin lesions in PULC-treated Nc/Nga mice. Moreover, PULC and its active component, astragalin, effectively increased the expression of filaggrin and involucrin in TNF-α/ IFN-γ-treated HaCaT cells. Taken together, PULC and astragalin have anti-AD effects.

## 5. Conclusions

Our results indicate that astragalin obtained from the leaves of *P. ussuriensis Maxim.* by bioactivity-guided isolation techniques had good activity in vitro. Topical application of PULC effectively reduced AD-like skin lesions by increasing skin thickening, reducing skin eczema and increasing skin hydration in DNCB-induced AD NC/Nga mice. Thus, PULC and astragalin may be an effective therapeutic agent, which may attenuate AD. Further studies are required to fully elucidate the underlying mechanism of *Pyrus ussuriensis Maxim.* leaves on AD. 

## Figures and Tables

**Figure 1 nutrients-11-00276-f001:**
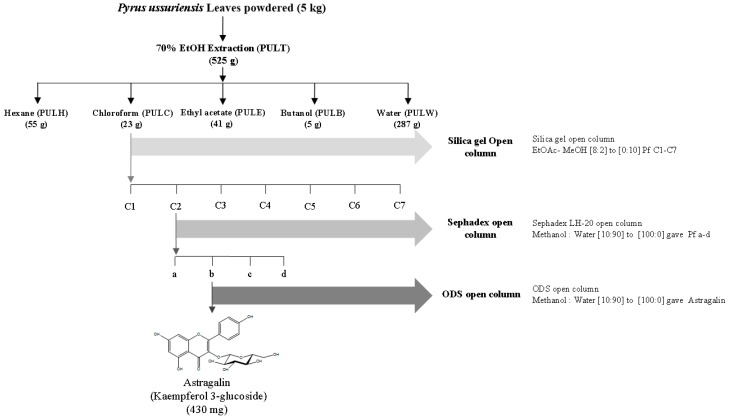
Extraction and isolation of bioactive compounds of from the chloroform (PULC) fraction using open column chromatography.

**Figure 2 nutrients-11-00276-f002:**
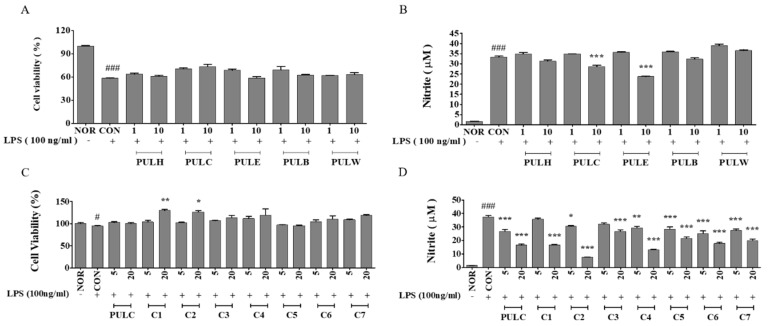
Effects of *P. ussuriensis Maxim* leaves (PUL) samples on cell viability (**A**,**C**) and NO production in RAW 264.7 macrophages treated with various doses of PUL and PULC fraction (**B**,**D**). NOR, untreated group; CON, LPS-induced group; PULH, hexane fraction treated group; PULC, chloroform fraction treated group; PULE, ethyl acetate fraction treated group; PULB n-butanol fraction treated group; PULW, water fraction treated group; C1-C7, PULC sub-fractions treated group. Results are expressed as mean ± standard error of the mean (SEM), (*N* = 3) # *p* < 0.05, ### *p* < 0.001 vs. NOR group; * *p* < 0.05, ** *p* < 0.01, and *** *p* < 0.001 vs. CON group.

**Figure 3 nutrients-11-00276-f003:**
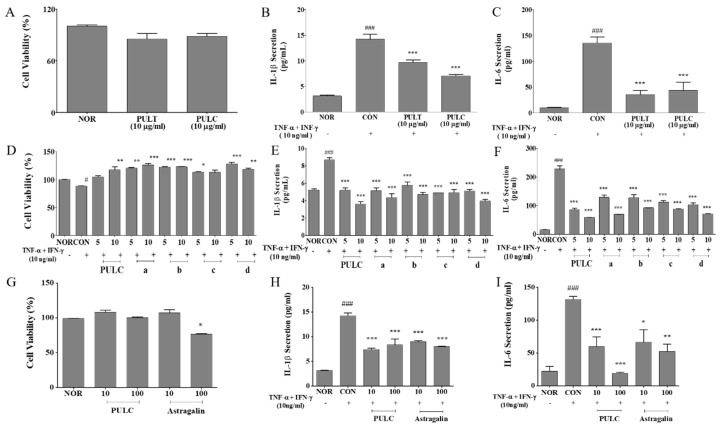
Effect of PUL samples on the cell viability (**A**,**D**,**G**), secretion of IL-1β and IL-6 in HaCaT cells treated with various doses of extract as indicated. Effect on TNF-α and IFN-γ-induced IL-β production (**B**,**E**,**H**). Effect on TNF-α and IFN-γ-induced IL-6 production (**C**,**F**,**I**). NOR, untreated group; CON, TNF-α and IFN-γ-induced group; PULT, total extract treated group; PULC, chloroform fraction treated group; a-d, sub-sub fractions of “C2” sub fractions treated group; Astragalin, isolated compound of “b” sub-sub fractions treated group. Results are expressed as mean ± SEM, (*N* = 3) # *p* < 0.05, ### *p* < 0.001 vs. NOR group; * *p* < 0.05, ** *p* < 0.01, and *** *p* < 0.001 vs. CON group.

**Figure 4 nutrients-11-00276-f004:**
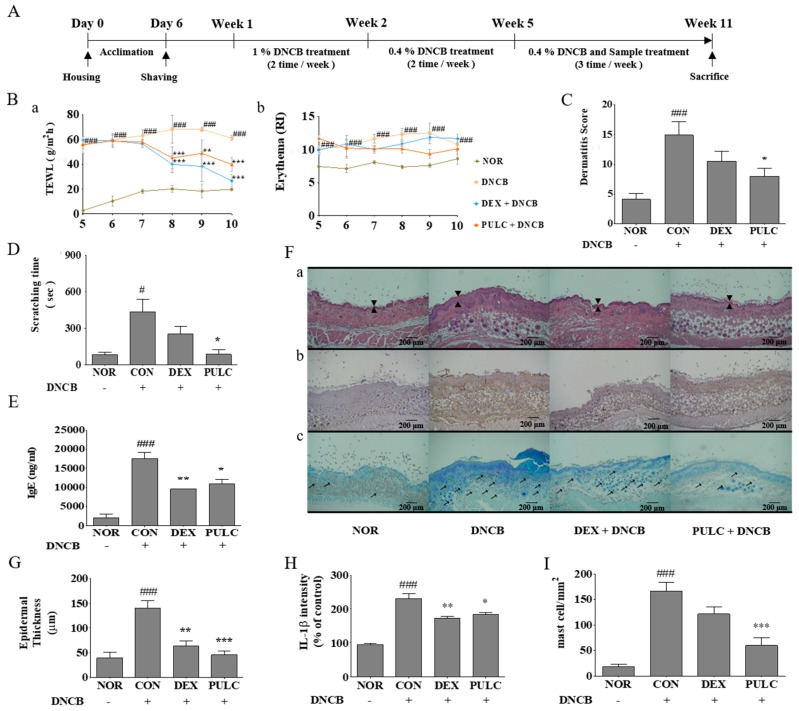
Experimental schedule and effects of PULC in DNCB-induced NC/Nga mice. (**A**) Experimental schedule for the induction of atopic dermatitis (AD) lesions; (**B**) Trans-epidermal water loss (TEWL) and erythema scores (**a**) TEWL (**b**) erythema scores; (**C**) Weekly clinical skin severity scores; (**D**) Scratching time on the last day of the experiment; (**E**) IgE levels evaluated on the last day of the experiment using enzyme-linked immunosorbent assay (ELISA); (**F**) Histological examination (**a**) H&E stain, arrows indicate thickness of the epidermis (**b**) IL-1β stain (**c**) toluidine blue stain, arrows indicate mast cells (scale bar = 200 μm); (**G**) Epidermal thickness; (**H**) IL-1β intensity; (**I**) Mast cell count. NOR, untreated group; CON, DNCB-treated groups; DEX, positive control (Dexamethasone 0.1%) treated group; PULC, PULC 0.5% treated group. Data represent the mean ± SEM. (*N* = 9) # *p* < 0.05, ### *p* < 0.001 vs. the NOR group; * *p* < 0.05, ** *p* < 0.01, and *** *p* < 0.001 vs. CON group.

**Figure 5 nutrients-11-00276-f005:**
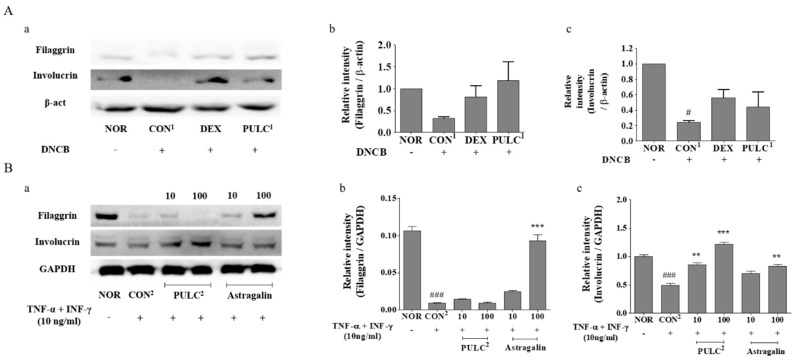
Effects of PULC and astragalin on protein expressions of filaggrin and involucrin. (**A**) Filaggrin and involucrin protein expressions in DNCB-induced NC/Nga mice (**a**) Filaggrin, involucrin and β-actin protein expressions (**b**) Relative intensity of filaggrin (**c**) Relative intensity of involucrin; (**B**) Filaggrin and involucrin protein contents in TNF-α and IFN-γ-induced HaCaT cells (**a**) Filaggrin, involucrin and GAPDH protein expressions (**b**) Relative intensity of filaggrin (**c**) Relativeintensityofinvolucrin; NOR, untreated group; CON^1^, DNCB-treated groups; DEX, positive control (Dexamethasone 0.1%) treated group; PULC^1^, PULC 0.5% treated group; CON^2^, TNF-α and IFN-γ-induced group; PULC^2^, chloroform fraction treated group. Data represent the mean ± SEM. (*N* = 3) # *p* < 0.05, ### *p* < 0.001 vs. NOR group; * *p* < 0.05, ** *p* < 0.01, and *** *p* < 0.001 vs. CON group.

**Table 1 nutrients-11-00276-t001:** Content of astragalin in the PULC fraction (Standard Rt: 20.53 ± 0.28, PULC Rt: 20.35 ± 0.17 min).

Content of Astragalin in the Pulc Fraction (Μg/100 Μg of Powdered Pulc Fraction)	
Sample	Astragalin
PULC	0.03 ± 0.00

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
