# Peer review of "Anti-Inflammatory Effect of Chloroform Fraction of Pyrus Ussuriensis Maxim. Leaf Extract on 2, 4-Dinitrochlorobenzene-Induced Atopic Dermatitis in nc/nga Mice"

_nutrients, 2019, doi:10.3390/nu11020276_

Round 1
Reviewer 1 Report
This is a well designed paper and the experimental approach is of high standards.
The authors are kindly asked to improve figures 2 and 3. The fonts are very small and the error bars are not visible. I would suggest that the filling of the bars is not black but a shade of grade; this would make the error bars visible.
Also, the different fractions used in figure 2 are not given their full name at the figure legend so the reader can not follow figure 2.
Also, the fonts in figures 2 and 3 are very small, almost unreadable.
Also, the introduction should include some info on inflammation and chronic diseases, suggested MS
https://www.mdpi.com/2072-6643/10/5/604
Atopic dermatitis is associated to inflammation, the authors need to address this.
I would suggest major revision.
Happy to review the revised MS.
Author Response
Response to Reviewer 1 Comments
Point 1:
The authors are kindly asked to improve figures 2 and 3. The fonts are very small and the error bars are not visible. I would suggest that the filling of the bars is not black but a shade of grade; this would make the error bars visible.
Also, the different fractions used in figure 2 are not given their full name at the figure legend so the reader can not follow figure 2.
Also, the fonts in figures 2 and 3 are very small, almost unreadable.
Point 2:
Also, the introduction should include some info on inflammation and chronic diseases, suggested MS
https://www.mdpi.com/2072-6643/10/5/604
Atopic dermatitis is associated to inflammation, the authors need to address this.
Response 1:
We thank reviewer for his/her valuable comments for this manuscript that helps us to improve the quality of our paper.
In the revised manuscript, we have improved all the figures (fonts size up, filling of the bars shade of grade, and given their full name at the figure legend).
-Line 216-222; 232-238; 252-260; 317-323
Response 2:
We have also added more information about atopic dermatitis and inflammation in the Introduction.
-Line 39 and 40 (Novak, N.; Bieber, T.; Leung, D.Y. Immune mechanisms leading to atopic dermatitis. The Journal of allergy and clinical immunology 2003, 112, S128-139, doi:10.1016/j.jaci.2003.09.032)
Reviewer 2 Report
This is interesting manuscript showing that PULC and astragalin ameliorate AD-like symptoms by suppressing both pro-inflammatory cytokines and immune stimuli in vitro and in animal models. The manuscript is well organized, contain literature adequate, is well written and provides interesting information for improving the treatment of AD. After few modification the manuscript can be improved. The major concern is the results and conclusion section.
The authors present the results in different order than figures. For example, the order of figure 4 in the text is 4E, 4C, 4B, 4D. Moreover, 4A and 4F are not included in the text. Similarly, figure 5A is missing in the text. The authors should fix it this section and present the results and figures in same order.
Figure legends are very poor, and should be improved. For example, Fig 2 what NOR, CON, etc. mean?. Similar to fig 3 with PULT. Moreover, fig 4 (line 231-234) is very confusing. Finally, fig 5 has a legend hard to understand, does not clear explication of its content, here authors also should clarify which results are in NC/Nga mice and which one in HaCaT cells.
The conclusion section did not support the results at all, it should be rewritten. Additional paragraph including limitations of the study should be added.
Editing of English language and style is required.
Author Response
Response to Reviewer 2 Comments
Point 1:
The authors present the results in different order than figures. For example, the order of figure 4 in the text is 4E, 4C, 4B, 4D. Moreover, 4A and 4F are not included in the text. Similarly, figure 5A is missing in the text. The authors should fix it this section and present the results and figures in same order.
Point 2:
Figure legends are very poor, and should be improved. For example, Fig 2 what NOR, CON, etc. mean?. Similar to fig 3 with PULT. Moreover, fig 4 (line 231-234) is very confusing. Finally, fig 5 has a legend hard to understand, does not clear explication of its content, here authors also should clarify which results are in NC/Nga mice and which one in HaCaT cells.
Point 3:
The conclusion section did not support the results at all, it should be rewritten. Additional paragraph including limitations of the study should be added.
Response 1:
We thank reviewer for raising this point. We truly agree and appreciate the suggestion by reviewer.
As suggested, we have changed the order of Fig 4 and have included all the full names in the Figure legends.
-Line 216-222; 232-238; 252-260; 317-323
Response 2:
In Fig5 legends we have also made the required changes. And carefully edited the figure legends to eliminate eventual remaining errors and discrepancies.
-Line 252-260; 77 and 78
Response 3:
We rewritten the conclusion about results of this study on 376-383 lines and the need for further study.
-Line 376-384
Round 2
Reviewer 1 Report
accept